# No Secondary Treatment Failure during Incobotulinumtoxin—A Long-Term Treatment Demonstrated by the Drawing of Disease Severity

**DOI:** 10.3390/toxins15070454

**Published:** 2023-07-12

**Authors:** Harald Hefter, Raphaela Brauns, Beyza Ürer, Dietmar Rosenthal, Philipp Albrecht, Sara Samadzadeh

**Affiliations:** 1Department of Neurology, Moorenstrasse 5, 40225 Düsseldorf, Germany; raphaela.brauns@uni-duesseldorf.de (R.B.); beyza.uerer@uni-duesseldorf.de (B.Ü.); dietmar.rosenthal@med.uni-duesseldorf.de (D.R.); philipp.albrecht@med.uni-duesseldorf.de (P.A.); sara.samadzadeh@yahoo.com (S.S.); 2Department of Neurology, Maria Hilf Clinics, 41063, Moenchengladbach, Germany; 3Charité—Universitätsmedizin Berlin, Corporate Member of Freie Universität Berlin and Humboldt-Universität zu Berlin, Experimental and Clinical Research Center, 13125 Berlin, Germany; 4Department of Regional Health Research and Molecular Medicine, University of Southern Denmark, 5230 Odense, Denmark; 5Department of Neurology, Slagelse Hospital, 4200 Slagelse, Denmark

**Keywords:** secondary treatment failure, primary treatment failure, focal dystonia, course of disease (CoD), CoD graphs, botulinum neurotoxin therapy, incoBoNT long-term treatment

## Abstract

The aim of this study was to detect clinical hints regarding the development of secondary treatment failure (STF) in patients with focal dystonia who were exclusively treated with incobotulinumtoxin/A (incoBoNT/A). In total, 33 outpatients (26 with idiopathic cervical dystonia, 4 with Meige syndrome and 3 with other cranial dystonia) who were treated with repeated injections of incoBoNT/A for a mean period of 6.4 years without interruptions were recruited to draw the course of their disease severity (CoD) from the onset of symptoms to the onset of BoNT therapy (CoDB graph) and from the onset of BoNT therapy to recruitment (CoDA graph). At the time of recruitment, the patients assessed the change in severity as a percentage of the severity at the onset of BoNT therapy. Blood samples were taken to test the presence of neutralizing antibodies (NABs) using the mouse hemidiaphragm assay (MHDA). Patients reported an improvement of about 70% with respect to the mean. None of the patients tested positive for MHDA. Three different types of CoDB and three different types of CoDA graphs could be distinguished. The patients with different CoDB graphs reported different long-term outcomes, but there was no significant difference in long-term outcomes between patients with different CoDA graphs. None of the patients produced a CoDA graph with an initial improvement and a secondary worsening as a hint for the development of STF. A primary non-response was not observed in any of the patients. During long-term treatment with BoNT/A, NABs and/or STF may develop. However, in the present study on patients with incoBoNT/A long-term monotherapy, no hints for the development of NABs or STF could be detected, underlining the low antigenicity of incoBoNT/A.

## 1. Introduction

Botulinum neurotoxins (BoNTs) are clostridial products produced as single chains [1] that have to be cleaved into a light and heavy chain coupled by a disulfide bond to become biologically active, and they are embedded in much larger protein complexes [1,2]. These complex proteins are necessary to protect BoNTs after oral uptake during their passage through the gastrointestinal tract against the acidic milieu of the stomach [2] and for the penetration of BoNTs through the gastrointestinal wall [3]. However, when BoNT preparations are injected, the complex proteins immediately start to diffuse from the neurotoxins after dilution [4,5] and may act as adjuvants [6,7,8,9] when dendritic cells are stimulated by transdermal applications [10].

In most long-term BoNT-treated patients, T-cells have recognized the presence of BoNT [11] and triggered the production of antibodies that may reduce the biological activity of BoNT (neutralizing antibodies (NABs)). NABs target the antigens of the heavy and the light chain [12,13] and reduce the intracellular uptake of BoNTs or the intracellular cleavage of SNARE-proteins [14,15]. This leads to the development of secondary treatment failure (STF). 

Clinically, the development of NAB-induced STF may be difficult to detect. It has been hypothesized that NAB-induced STF occurs early in the course of treatment [16]. For several indications, the development of STF after the first three injections has been reported [17,18]. First, the duration of the clinical effect is reduced [19]. During this phase of the development of NAB-induced STF, the reduction in efficacy can be overcome by an increase in the dose (partial STF (pSTF)). With the continuation of BoNT applications, NAB titers usually increase until a complete STF (cSTF) develops with a large reduction in the clinical effect, including the peak effect around week 4 [17,19]. Treating physicians and patients often realize that BoNT injections have become less effective compared to the effect at the onset of BoNT therapy only after years of treatment [16].

When STF (pSTF or cSTF) is suspected and NABs are determined, some patients may have a negative NAB test but develop a clear STF [20,21]. On the other hand, the cross-sectional testing of NABs in still-responding patients may detect patients with excellent clinical outcomes but positive NAB testing [22,23]. This raises the question of the sensitivity and specificity of positive NAB tests for STF. It may very well be that careful clinical observation and documentation are at least as sensitive as the performance of a mouse lethality assay (MLA) or the mouse hemidiaphragm test (MHDA).

We, therefore, developed a method of drawing the course of disease severity (CoD). This method has the advantage of using the patient’s assessment of the clinical efficacy of BoNT therapy, which may add information to the treating physician’s assessment [24,25,26]. Furthermore, this drawing test is much cheaper and easier to perform than a laboratory NAB test. In a cohort of 66 consecutively recruited patients with cervical dystonia, 17 patients (=25.8%) produced a CoD graph from the onset of BoNT/A therapy to recruitment (CoDA graph) that showed a good initial response followed by secondary worsening. This secondary worsening was also observed by the treating physician who previously increased the BoNT/A dose and optimized the injection technique. In ten of these seventeen patients (=58.8%), a positive MHDA test was detected [25,26]. This may indicate that the clinical situation (especially the CoD drawing) is at least as sensitive to detecting STF as MHDA testing is to detecting NAB-induced STF.

To present more evidence that CoD drawing is a useful clinical tool in detecting STF and does not produce false-positive cases with regard to STF, CoD drawing was compared to MHDA testing in 33 patients with focal cranial dystonia who were treated long-term with incoBoNT/A exclusively and for whom no NAB induction was expected [27]. 

## 2. Results

### 2.1. Demographical, Treatment-Related Data and Outcome Measures of All Patients

The final analysis, which is presented in detail in Table 1, was based on 33 patients (17 females and 16 males). The mean age (AGE) was 58.7 years (SD: 13.1), and the mean age at onset of symptoms (AOS) was 47.6 years (SD: 15.0). The time to therapy (DURS) was 5.2 years (SD: 8.1), and the duration of treatment (DURT) 6.4 years (SD: 2.2). 

The initial dose (IDOSE) was 164 U incoBoNT/A (SD: 85). The dose was increased (INDOSE) during DURT by 96 U incoBoNT/A with respect to the mean (SD: 80). The dose at recruitment (ADOSE) was therefore 260 U (SD: 111).

The improvement assessed by the patients as a percentage of the initial severity of their disease (IMPQ) was 69.4% (SD: 19). When patients had to use the right edge of the CoDA graph as a VAS scale (IMPD) to assess the actual severity of the disease (ASD) before drawing the CoDA graph (for details, see methods), the improvement (IMPD) was 69.6% (SD: 18.2), which was nearly identical to the IMPQ.

In the patients with CD (n = 26), the initial TSUI score varied between 4 and 16 and was 7.6 with respect to the mean (SD: 3.0). At recruitment, the mean TSUI score (ATSUI) was 4.0 (SD: 3.0). Thus, the mean improvement in the TSUI score (IMPTSUI) was 3.6 = 47.6% (SD: 3.5).

### 2.2. Three Types of CoDB Graphs and Corresponding Treatment-Related Data

Three different types of CoDB graphs could be distinguished. Seven patients experienced a rapid onset of their disease with little further progression thereafter. These patients produced CoDB graphs with more than 75% of the data points lying above the line (OS/OT-line) from the left lower corner (onset of symptoms) to the right upper corner (onset of therapy) (for further details, see methods). This type of CoDB graph was called the rapid onset (RO) type. Eleven patients experienced a slow, mild progression of their disease for a longer period of time before their disease started to worsen more rapidly. These patients produced CoDB graphs with more than 75% of the data points lying below the OS/OT-line. This type of CoDB graph was called the delayed onset (DO) type. Fifteen patients experienced a continuous worsening of disease severity from the onset of symptoms to the onset of incoBoNT/A therapy. Their CoDB graphs varied around the OS/OT line. This type of CoDB graph was called the continuous onset type (CO-type). 

The outcome assessed by the patients (IMPQ) was significantly different across the three RO, CO and DO subgroups. The best outcome at recruitment (IMPQ: MV = 77.9% (SD: 8.1)) was reported by the RO group, and the lowest (MV = 57.0%; SD: 23.2%) was reported by the DO group (see Table 1). The initial dose (ITSUI; *p* < 0.009) and actual dose per session (ATSUI; *p* < 0.01)) were also significantly different across the three patient subgroups. The lowest IDOSE (MV = 105; SD = 72) and ADOSE (MV = 199; SD = 108) by far were used in the CO group (see Table 1 and Figure 1).

In CD patients (n = 26), TSUI scores were also determined by the treating physician. The best outcome (ATSUI: MV = 3.0; SD = 1.6) was observed in the RO group, and the worst outcome was observed in the DO group (MV = 5.0; SD = 4.4). This explains the significant difference in the patient assessment of improvement (IMPQ: see Table 1).

In the RO group, the TSUI score decreased from a high initial TSUI (MV = 9.7) to the best TSUI (BTSUI) of 1.9. Thereafter, the TSUI score increased again to the actual TSUI (difference: 3.0 − 1.9 = 1.1). This worsening was compensated for by an increase in dose (INDOSE) of 40 U (ADOSE − BDOSE = 314 − 274). In the CO group, the TSUI score worsened from the best TSUI of 0.9 to 3.8 (difference: 2.9), which was only partially compensated for by an increase in the dose of 14 U (ADOSE − BDOSE = 199 − 185). In the DO group, the TSUI score worsened from 1.8 to 5.0 (difference: 3.2), which was not compensated for by an increase in the dose (ADOSE − BDOSE = 310 − 302 = 8). Patient assessment (IMPQ) correlates with the quality of the compensation strategy being performed by the treating physician.

### 2.3. Three Different Types of CoDA Graphs and Corresponding Treatment-Related Data

Three different types of CoDA graphs could be distinguished. To classify the CoDA graphs, a line (OT/ASD line) between the left upper corner (onset of therapy) and the mark for the actual severity of the disease (ASD mark on the right edge of the CoDA graph) was drawn. Nine patients drew CoDA graphs with more than 75% of the data points lying above the OT/ASD line (for further details see methods). This type of CoDA graph was called the rapid response (RR) type. Six patients experienced a slow initial improvement of their disease for a longer period of time before their disease started to improve more rapidly. These patients produced CoDA graphs with more than 75% of the data points lying below the OT/ASD line. This type of CoDA graph was called the delayed response (DR) type. Eighteen patients experienced a continuous improvement in disease severity from incoBoNT/A therapy to recruitment. Their CoDA graphs varied around the OT/ASD line. This type of CoDA graph was called the continuous response (CR) type. 

When the entire cohort was split up into three subgroups according to response type, the female/male ratio in the different subgroups was different (p = 0.05; chi2-testing). The three-group ANOVA did not reveal any significant differences between the RR, CR or DR subgroups. AGE (MV = 67.2; SD = 9.8) was the highest in the DR group, as was AOS (MV = 52.5; SD: 8.4). This group also had the longest duration of treatment (MV = 7.8 years; SD = 3.0 years). Although the initial TSUI score was the highest (MV = 10.0; SD = 6.0) in the DR group, the actual TSUI score (ATSUI) was the lowest (MV = 2.7; SD = 1.2). This was achieved using the highest increase in dose (INDOSE) from 132 to 249 (difference = 117) (see Figure 2).

### 2.4. No NAB Induction and No Primary or Secondary Treatment Failure

In all 33 patients, the time to paralysis (TTP) of the MHDA test was well below 60 min (see Table 2). Thus, all patients had a negative MHDA test. All patients experienced an improvement of at least 40%. No CoDA graph yielded a hint that a patient was a primary non-responder. Furthermore, no patient produced a graph with an initial improvement of more than 20% followed by secondary worsening. No CoDA graph yielded a hint that a patient had developed a partial or complete secondary treatment failure (STF).

## 3. Discussion

### 3.1. No NAB Induction during Long-Term Monotherapy with incoBoNT/A

The cross-sectional testing of the present cohort for the presence of NABs by means of the sensitive MHDA did not yield any positive results as hypothesized in the Introduction. In all patients, the time to paralysis (TTP) did not exceed 55 min and was well below the lower cut-off value for an MHDA-positive test result of 60 min. This is in line with previous reports on NABs in patients exclusively treated with incoBoNT/A [27]. So far, no patient has been detected who developed NABs after incoBoNT/A monotherapy. However, a few preceding injections with a complex protein containing BoNT/A preparation before long-term incoBoNT/A therapy may themselves induce NAB formation, so the risk of developing NABs is not always zero [28].

In patients being exclusively treated with a complex protein containing BoNT/A preparation, the incidence of becoming MHDA positive was about 1.1% [27]. After a mean duration of therapy of 6.4 years (as in the present cohort), a prevalence of at least 7.2% of NAB-positive patients must be expected if the antigenicity of incoBoNT/A was equal to that of abo- or onaBoNT/A. This implies that at least two MHDA-positive patients were expected in the present cohort. However, no patient was MHDA-positive. This is consistent with the observation in a much larger study [27] that the antigenicity of incoBoNT/A is significantly lower than that of abo- or onaBoNT/A.

### 3.2. No Hints for the Development of a Secondary Treatment Failure in the CoDA Graphs

NAB testing was compared to the CoD graph drawing. No patient (with CD or with another cranial dystonia) produced a CoDA graph with an initial improvement followed by a secondary worsening (STF-type) as a hint for an STF.

In a previous study on a patient drawing of the course of CD severity after the onset of botulinum toxin therapy (CoDA graph), 17 (=25.8%) out of 66 patients (who were consecutively recruited and mainly treated with abobotulinumtoxinA (aboBoNT/A; Dysport^®^) or onabotulinumtoxinA (onaBoNT/A; Botox^®^)) produced CoDA graphs of an STF type. In five of these patients, STF was complete (cSTF: 5/66 = 7.6%), which is slightly more than the 5.9% reported by Walter et al. [17]. This may be a hint that the risk to develop a partial or complete secondary treatment failure under incoBoNT/A therapy is lower than under abo- or onaBoNT/A therapy, as was reported for the risk of NAB induction [27]. 

The incidence of the development of a pSTF appears to be at least twice as high as the incidence of NAB induction (incidence of NAB induction = 1.1%; incidence of STF development = 2.6%) [20,25]. This indicates that the careful clinical assessment of the efficacy of BoNT/A therapy may be superior to laboratory tests in detecting the development of STF. In the present study, neither the patient assessment of the clinical effect by means of a CoD graph drawing nor physician clinical scoring detected STF, underlining the low antigenicity of incoBoNT/A. This supports the notion that CoD graph drawing may be a useful clinical tool for detecting STF.

It has to be kept in mind that “secondary treatment failure” is not exactly defined. It has repeatedly been reported that in some patients with an STF change in injection guidance, the modification of the injection scheme and increase in dose led to improved response behavior. We think that all these cases of STF that can be improved by the modification of the BoNT/A injection technique or dose should be classified under insufficiently treated patients instead of patient treatment failures. 

This insufficiency of treatment may result from a further progression of the underlying neurological disease despite improvements under ongoing BoNT/A therapy [29]. In patients with focal dystonia, a focal type may worsen to a multifocal or segmental type, affording the adaptation of the injection scheme and dose. In the present study, a worsening in disease severity is documented in patients with CD. After the best outcome (BTSUI) was reached under ongoing incoBoNT/A treatment, the TSUI score slightly worsened again. The treating physician reacted to this development of the TSUI score by adapting the dose. In the case of too-low compensation, the patients reported a reduced efficacy of incoBoNT/A therapy, as in the DO group (see Results). This underlines the necessity to continuously adapt the dose to clinical needs and to distinguish between treatment failure and insufficient treatment.

### 3.3. No Hint of a Primary Treatment Failure

In the previously mentioned report on CoDA graph drawing [24,25], a subgroup was detected with a poor response during the entire duration of treatment (PR subgroup). These 12 (18%) out of 66 patients were classified as primary non-responders, had the lowest improvement in TSUI score (mean IMPTSUI = 1.6), and reported the least improvement (mean IMPQ = 11.3%; mean IMPD = 7%), and they had the longest time to therapy (mean DURS = 93 months = 7.75 years) and the shortest duration of treatment (mean DURT = 75 months = 6.25 years). The obvious reason is that adherence to therapy is low when the efficacy of the BoNT treatment is low [30].

In the present study, no poor or primary non-responders could be detected. The initial response was low in the DR group. The simple reason for this is that the initial dose of 132 U incoBoNT/A with respect to the mean was too low. With a considerable increase in the dose (INDOSE) by 118 U, the best mean outcome (BTSUI) could be achieved. This further emphasizes the necessity for the adaptation of the dose to clinical needs. In a previous paper, we emphasized that in patients with fairly high initial severity (as in the DR subgroup of the present cohort), an unusually high dose may be necessary at the beginning of BoNT therapy to achieve a satisfying response [31]. If this adaptation is not performed, the risk is high that the patient terminates BoNT therapy and is classified as a primary non-responder [31].

### 3.4. Strengths and Limitations of the Study

The strength of the study is that antibody testing, patient assessment of efficacy, and the treating physician scoring of the treatment effect all confirm the low antigenicity of incoBoNT/A, and there is evidence showing that CoD graph drawing may be a useful clinical tool. The size of the present cohort is small (n = 33), but it is large enough that MHDA-positive patients or patients with an STF could have been detected if the incidences for NAB induction or STF development under incoBoNT/A therapy were equally high as under abo- or onaBoNT/A therapy.

For the present study, a mixed population of patients with CD and other forms of cranial dystonia was recruited. However, the small number of patients did not allow us to analyze differences between patient subgroups, although the course of the disease and response to BoNT/A may have been different.

Furthermore, CoD graphs were produced on the day of recruitment and were based on the patient’s rcall. Therefore, a longitudinal study is recommended for testing CoD graph drawing repeatedly at short and long time intervals to control for recall bias.

We recommend that a multi-center, long-term, prospective study be performed in order to analyze the patient’s assessment of the efficacy of BoNT/A therapy using CoD graph drawing, the treating physician’s assessment of the efficacy of BoNT/A therapy by means of clinical scales and the risk of antibody formation by testing patients repeatedly for the presence of NABs. We are aware that this will be a challenging but nevertheless necessary study to more clearly demonstrate differences in the antigenicity of different BoNT/A preparations and long-term outcomes.

## 4. Conclusions

The patient assessment of the clinical effect of BoNT therapy and treating physician scoring both demonstrate that the risk of antibody formation and the development of a secondary treatment failure are low under continuous incoBoNT/A therapy. An analysis of patient drawings of the course of disease severity may yield helpful information for the treating physician to optimize BoNT/A therapy.

## 5. Materials and Methods

The present study was performed according to the guidelines for good clinical practice (GCP) and according to the Declaration of Helsinki. It was approved by the local ethics committee of the University of Düsseldorf (number: 4085).

### 5.1. Patients: Demographical and Treatment-Related Data

The inclusion criteria of the study were (i) age over 17; (ii) diagnosis of idiopathic cervical dystonia (CD), Meige syndrome or oromandibular dystonia; (iii) onset of therapy in the outpatient department of the University of Düsseldorf (Germany) and continuous treatment every 12 to 13 weeks without the interruption of BoNT therapy of more than one treatment cycle; (iv) at least three injections of BoNT; and (v) treatment with incoBoNT/A only. Exclusion criteria were (i) patients under legal care; (ii) patients with multifocal or segmental dystonia; (iii) additional disabling disease other than CD; and (iv) patients with a clinical overt disturbance of mood or perception.

More than 500 charts of regularly treated patients in the outpatient department of the Neurological Clinic of the University hospital in Düsseldorf (Germany) were screened for interruptions in ongoing BoNT therapy. Patients with interruptions of less than two treatment cycles who fulfilled all inclusion criteria but no exclusion criteria were informed of the purpose of the study while waiting in the outpatient department for their next injection. In total, 35 patients (26 with CD and 9 with other cranial dystonia) who gave informed written consent were consecutively recruited. 

The following demographic data (age at day of recruitment (AGE), age at onset of symptoms (AOS), age at onset of therapy (AOT), duration of therapy (DURT)) and the following treatment-related data (initial total dose (IDOSE) and actual total dose (ADOSE)) were extracted from the charts of the patients. The time to therapy (DURS = time span during which patients tolerated symptoms without BoNT therapy) was calculated as DURS = AOT − AOS. The increase in dose (INDOSE) during treatment was calculated as ADOSE-IDOSE. 

Patients were asked to rate the change in the severity of their disease since the onset of BoNT therapy in percent of the severity at the onset of therapy (IMPQ, improvement (plus value) and worsening (minus value)). 

In patients with CD, the treating physician scored the actual severity of CD on the day of recruitment by means of the TSUI score [32] (ATSUI). The initial TSUI score at the onset of therapy (ITSUI) was extracted from the chart of the patient, and improvement in the severity of CD determined by means of the TSUI score (IMPTSUI) was calculated as (ITSUI − ATSUI) × 100/ITSUI). The best TSUI score (BTSUI) during incoBoNT/A therapy was also extracted from the chart as well as the corresponding dose (BDOSE) at the time BTSUI occurred. 

### 5.2. Drawing of the Course of Disease Graphs

The courses of disease before and after the onset of BoNT therapy were recalled by the patient on the day of recruitment.

The drawing of the course of disease graphs (CoD graphs) has been described in detail previously [24,25]. The patients were comfortably seated in front of a desk; one hand held a square piece of paper with a size of 10 × 10 cm, and the other held a pen. For the drawing of the CoD graph before BoNT therapy (CoDB graph), patients were instructed to draw a continuous graph representing their CoD severity from the onset of symptoms to the onset of BoNT/A therapy by starting at 0 at the left lower corner (onset of symptoms) and ending at the right upper corner (100% = severity of CD at the onset of BoNT/A therapy). Thereafter, a second square piece of paper measuring 10 × 10 cm in size was presented to the patient. Before drawing the CoD graph from the onset of BoNT therapy to the day of recruitment (CoDA graph), patients had to mark the actual severity of their disease on the day of recruitment (ASD) on a line running through the right edge of the square. Then, they were instructed to draw a continuous graph into a second square (representing their CoD severity from the onset of BoNT/A therapy until the actual day of recruitment) starting from the left upper corner (=100% = severity at the onset of BoNT/A therapy) to the ASD mark on the right edge of the square. Three attempts for CoDB and CoDA graph drawing were allowed, as well as verbal help by the investigator, but drawing assistance was not allowed. No example of a CoD graph was shown to avoid any bias. The improvement estimated by drawing the CoDA graph (IMPD) was calculated as (10 − ASD) × 10.

After the drawing of the CoD graphs, patients underwent a detailed clinical investigation and received their routine incoBoNT/A-injection. Thereafter, a blood sample was taken, centrifuged and deeply frozen for later NAB testing.

### 5.3. Classification of the CoD Graphs

Two of the thirty-five patients were unable to draw a continuous CoDB and/or CoDA graph. Thirty-three CoDB and CoDA graphs were scanned by means of a standard scanner. The commercially available software DIGITIZEIT^®^ [33] was used to digitize the graphs. The origin and end of the *x*-axis, the origin and end of the *y*-axis and the origin and end of the CoD graphs were necessarily marked on the scan. The software was used to produce an x,y-table for a graph when a stick was moved along the scan from the origin to the end of the graph. These data were used to produce the digitized version of the graph, which was used for further analysis. 

The classification of CoDB graphs into three different types depending on the main curvature has been described previously [24,25]. When more than 75% of the CoDB graph was drawn above the 45-degree line from the left lower to the right upper corner, the graph was classified as a “rapid onset” (RO) type. When the graph oscillated around the 45-degree line, the graph was classified as a “continuous onset” (CO) type. When 75% of the graph was drawn below the 45-degree line, the graph was classified as a “delayed onset” (DO) type.

The classification of the CoDA graphs was performed similarly. In the first step, a line between the left upper corner, and the ASD mark was drawn. A graph that was drawn more than 75% below this line was classified as a “rapid response” (RR) type, a graph that closely followed this line was classified as a “continuous response” (CR) type and a graph that was drawn 75% above this line was classified as a “delayed response” (DR) type. 

The CoD graphs were independently classified by both authors: RB and BÜ. In five discordant cases, the final classification was achieved during a consensus meeting. 

### 5.4. Antibody Testing

After the recruitment of all patients, pseudorandomized coded blood samples were sent to the Toxogen^®^ laboratory (MMH, University of Hannover, Hanover, Germany) (without any information about clinical data) to be tested for the presence of NABs by means of the MHDA. All 33 blood samples were tested within one batch. The Toxogen^®^ laboratory determined the time to paralysis (TTP) for each of the 33 blood samples and decided whether the blood sample was MHDA-positive or not. 

### 5.5. Statistics

The final data analysis was based on 33 patients (26 patients with CD, 4 patients with Meige syndrome and 3 patients with oromandibular or oropharyngeal dystonia) with a complete data set.

The patients were split up into three subgroups (RO, CO and DO subgroups) according to the CoDB graphs. A chi2 analysis was performed for whether sex distribution was different across patient subgroups. A three-group ANOVA was performed to test whether AGE, AOS, DURS, DURT, IDOSE, ADOSE, INCD, ITSUI, ATSUI, IMPTSUI, IMPQ, IMPD, BDOSE, BTSUI and TTP were significantly different between the RO, CO and DO subgroups (see Table 1).

The patients were also split up into three other subgroups (RR, CR and DR subgroup) depending on the CoDA graph type. Again, a chi2 analysis was performed for whether sex distribution was different across patient subgroups. A three-group ANOVA was performed to test whether AGE, AOS, DURS, DURT, IDOSE, ADOSE, INCD, ITSUI, ATSUI, IMPTSUI, IMPQ, IMPD, BDOSE, BTSUI and TTP were significantly different between the RR, CR and DR subgroup (see Table 2). All statistical analyses were performed using the SPSS^®^ statistics package (version 25; IBM, Armonk, NY, USA).

## Figures and Tables

**Figure 1 toxins-15-00454-f001:**
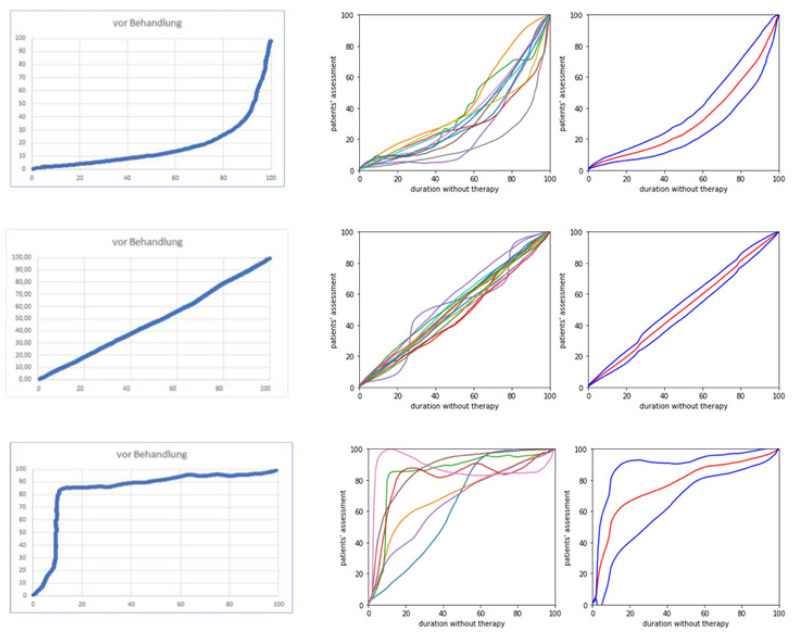
For each of the three patient subgroups (DO subgroup (**upper row**), CO subgroup (**middle row**) and RO subgroup (**lower row**)) a typical CoDB graph is demonstrated on the left side; all CoDB graphs in the middle part and the average CoDB graph and the corresponding plus/minus 1 standard deviation range across all patients within the subgroup are on the right side.

**Figure 2 toxins-15-00454-f002:**
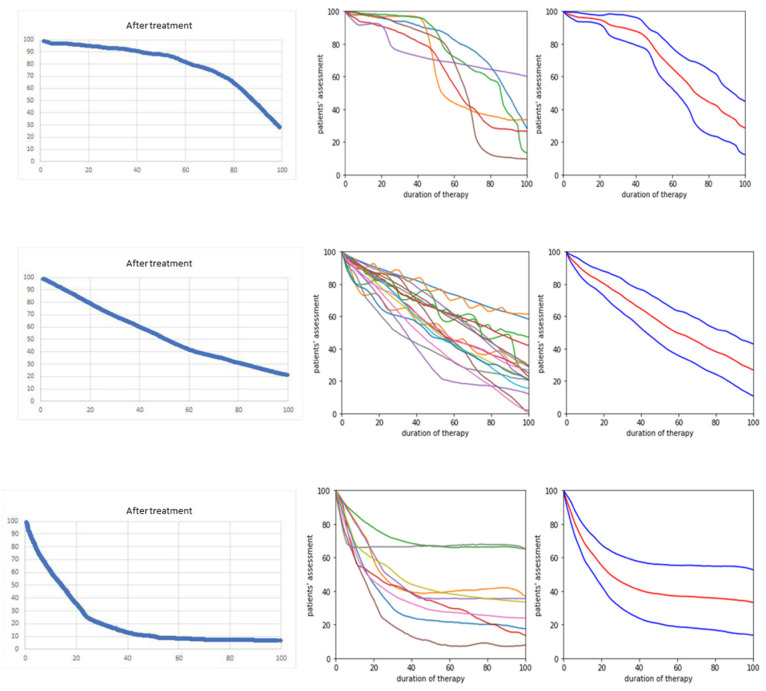
For each of the three patient subgroups (DR subgroup (**upper row**), CR subgroup (**middle row**) and RR subgroup (**lower row**)), a typical CoDA graph is demonstrated on the left side; all CoDA graphs in the middle part and the average CoDA graph and the corresponding plus/minus 1 standard deviation range across all patients within the subgroup are on the right side. No patient had a CoDA graph exhibiting initial improvement and secondary worsening, indicating a secondary treatment failure.

**Table 1 toxins-15-00454-t001:** Statistical comparison (three-group ANOVA) of demographic and treatment-related data between the RO, CO and DO subgroups.

Parameter		RO	CO	DO	ALL	Significance-Level (*p* < *0*.05)
n =		7	15	11	33	
Female/male		¾	10/5	4/7	17/16	n.s.
AGE (years)	MV/SD	53.9/14.1	59.67/13.13	60.5/13.1	58.72/13.15	n.s.
	MIN–MAX	28.62–71.58	34.76–82.22	30.35–81.44	28.62–82.22	
AOS (years)	MV/SD	43.4/17.6	46.46/14.20	51.4/15.0	47.57/15.04	n.s.
	MIN–MAX	15.89–63.01	13.18–67.03	22.38–73.89	13.18–73.89	
DURS (months)	MV/SD	67.2/139.2	73.08/108.36	42/52.8	62.04/96.96	n.s.
	MIN–MAX	1.2–336.00	3.6–336.00	1.2–192.00	1.2–336.00	
DURT (months)	MV/SD	63.36/21.48	84/29.64	72/24	76.32/25.29	n.s.
	MIN–MAX	48.00–108.00	48.00–132.00	48.00–108.00	48.00–132.00	
IDOSE	MV/SD	202.86/69.63	105.50/71.95	218.64/58.66	163.86/84.91	*p* < 0.009
	MIN–MAX	100.00–300.00	30.00–230.00	80.00–300.00	30.00–300.00	
ADOSE	MV/SD	314.29/69.01	198.67/108.31	309.55/100.56	260.15/111.52	*p* < 0.01
	MIN–MAX	200.00–400.00	35.00–400.00	160.00–500.00	35.00– 500.00	
INDOSE	MV/SD	111.43/23.22	93.17/77.91	90.91/99.64	96.29/76.97	n.s.
	MIN–MAX	80.00–150.00	0.00–300.00	−40.00–260.00	−40.00–300.00	
ITSUI	MV/SD	9.67/3.67	5.57/2.15	7.75/2.96	7.57/3.25	n.s.
	MIN–MAX	6.00–16.00	4.00–10.00	4.00–12.00	4.00–16.00	
ATSUI	MV/SD	3.00/1.55	3.75/1.67	5.00/4.42	3.96/2.99	n.s.
	MIN–MAX	1.00–5.00	2.00–7.00	0.00–12.00	0.00–12.00	
IMPTSUI	MV/SD	6.67/3.88	1.86/2.73	3.63/5.13	3.60/3.5	n.s.
	MIN–MAX	1.00–12.00	−1.00–7.00	−4.00–12.00	−4.00–12.00	
IMPQ	MV/SD	77.86/8.09	73.73/16.07	57.00/23.24	69.41/19.00	*p* < 0.03
	MIN–MAX	65.00 -| 90.00	40.00–96.00	0.00–90.00	0.00–96.00	
IMPD	MV/SD	73.54/21.88	71.44/15.78	64.58/19.57	69.60/18.22	n.s.
	MIN–MAX	29.60–100.00	35.20–88.00	33.8–98.60	29.60–100.00	
BDOSE	MV/SD	274.29/95.54	185.00/70.91	301.67/73.40	249.42/92.42	*p* < 0.01
	MIN-MAX	170.00–400.00	60.00–300.00	200.00–400.00	60.00–400.00	
BTSUI	MV/SD	1.86/1.68	0.89/1.17	1.78/1.79	1.48/1.56	n.s.
	MIN–MAX	0.00–4.00	0.00–3.00	0.00–5.00	0.00–5.00	
TTP (min)	MV/SD	48.86/3.13	48.54/3.55	46.00/2.49	47.71/3.28	n.s.
	MIN–MAX	44.00–52.00	43.00–55.00	41.00–51.00	41.00–55.00	

The abbreviations of the parameters in column 1 are explained in the list of abbreviations and the methods. MV: Mean value; SD: standard deviation; MIN: minimum in the patient group; MAX: maximum in the patient group.

**Table 2 toxins-15-00454-t002:** Statistical comparison (three-group ANOVA) of demographic and treatment-related data between the RR, CR and DR subgroups.

Parameter		RR	CR	DR	ALL	Significance-Level (*p* < 0.05)
N =		9	18	6	33	
Female/male		2/7	12/6	4/2	18/15	*p* = 0.05
AGE (years)	MV/SD	60.0/11.32	55.27/14.09	67.17/9.79	58.72/13.15	n.s.
	MIN–MAX	44.87–76.57	28.62–81.44	56.1–82.22	28.62–82.22	
AOS (years)	MV/SD	47.56/17.63	45.94/15.72	52.5/8.41	47.57/15.03	n.s.
	MIN–MAX	13.18–64.93	15.89–72.89	41.75–67.30	13.18–72.89	
DURS (months)	MV/SD	88.2/112.8	44.8/76.08	73.8/120.96	61.2/95.88	n.s.
	MIN–MAX	8.04–323.64	1.2–338.76	6.00–343.44	1.2–343.44	
DURT (months)	MV/SD	74.76/23.28	44.8/21.24	93.6/36	76.32/25.92	n.s.
	MIN–MAX	51.6–110.4	48.0–124.8	45.6–142.8	45.6–142.8	
IDOSE	MV/SD	140.83/92.94	186.11/76.50	131.67/91.91	163.86/84.91	n.s.
	MIN–MAX	30.00–300.00	30.00–300.00	35.00–275.00	30.00–300.00	
ADOSE	MV/SD	215.0/130.0	286.4/89.53	249.17/137.86	260.15/111.6	n.s.
	MIN–MAX	35.00–400.00	100.00–500.00	80.00–400.00	35.00–500.00	
INDOSE	MV/SD	74.17/70.47	100.27/72.52	117.5/102.99	96.29/76.97	n.s.
	MIN–MAX	0.00–200.00	−40.00–260.00	0.00–300.00	−40.00– 300.00	
ITSUI	MV/SD	6.20/2.28	7.54/2.75	10.0/6.0	7.6/3.25	n.s.
	MIN–MAX	4.00–9.00	4.00–12.00	4.00–16.00	4.00–16.00	
ATSUI	MV/SD	4.00/2.12	3.77/3.06	2.67/1.15	3.96/2.99	n.s.
	MIN–MAX	2.00–7.00	0.00–12.00	2.00–4.00	0.00–12.00	
IMPTSUI	MV/SD	1.25/3.49	3.77/4,7	7.33/5.03	3.9/4.36	0.24; n.s.
	MIN–MAX	1.00–2.00	−4.00–12.00	2.00–12.00	−4.00–12.00	
IMPQ	MV/SD	70.11/29.9	67.06/11.18	75.0/18.71	69.4/18.99	n.s.
	MIN–MAX	60.00–96.00	0.00–80.00	40.00–90.00	0.00–96.00	
IMPD	MV/SD	67.61	70.52/16.44	69.82/19.49	69.6/18.22	n.s.
	MIN–MAX	29.60–91.50	33.80–100.00	35.20–88.00	−29.6–100.00	
BDOSE	MV/SD	218.57/106.53	246.79/88.1	257.50/117.86	249.42/92.42	n.s.
	MIN-MAX	60.00–400.00	100.00–400.00	130.00–400.00	60.00–400.00	
BTSUI	MV/SD	1.43/1.81	1.26/1.49	2.67/1.15	1.68/1.8	n.s.
	MIN-MAX	0.00–5.00	0.00–4.00	0.00–4.00	0.00–5.00	
TTP (min)	MV/SD	48.56/3.32	47.7/3.41	46.20/2.77	47.71/3.28	n.s.
	MIN–MAX	44.00–55.00	41.00–52.00	44.00–51.00	41.00–55.00	

Abbreviations of the parameters in column 1 are explained in the list of abbreviations and in the Materials and Methods Section. MV: Mean value; SD: standard deviation; MIN: minimum in the patient group; MAX: maximum in the patient group.

## Data Availability

Data are available upon request related to the restrictions of privacy or ethics. The data presented in this study are available upon request from the corresponding author.

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
