# Peer review of "No Secondary Treatment Failure during Incobotulinumtoxin—A Long-Term Treatment Demonstrated by the Drawing of Disease Severity"

_toxins, 2023, doi:10.3390/toxins15070454_

Round 1
Reviewer 1 Report
In this manuscript, the authors conducted a study using patient reported trends in disease progression and response to botulinum toxin treatment in addition to testing for neutralizing antibodies and physician impressions of response to determine whether there was evidence of secondary treatment failure in participants with focal dystonia (cervical and/or cranial dystonia). Participants were asked to draw a curve of their disease progression prior to starting treatment with botulinum toxin and a second curve of their progression after initiation of treatment with botulinum toxin. Blood samples taken from each participant at the time of enrollment were tested for the presence of neutralizing antibodies to botulinum toxin.
Patient drawn graphs separated into 3 distinct types/categories before botulinum toxin treatment initiation and 3 types/categories after initiation of botulinum toxin. None of the patient samples tested positive for neutralizing antibodies. This was consistent with results on patient drawn graphs of treatment response, which did not show occurrences of either primary or secondary non-response. This paper demonstrates how patient reported measures such as graphing of disease severity can be helpful in demonstrating disease progression and response to treatment.
The paper could be strengthened with some clarifications and further revision.
Introduction:
· Line 65: Sentence “Clinically, the development of NAB-induced may be difficult to detect.” Appears to be missing a word. I believe you meant to say NAB-induced STF
· Lines 86-91 the authors present data from another study that demonstrate that more patients reported secondary clinical worsening via CoD graphs than those that had positive MHDA testing. This point is critical, as it serves as the basis for the authors suggestion that their method of drawing the course of disease severity could be more sensitive than MHDA testing in detection of NAB-induced STF and why the CoD graphs were a useful tool to detect evidence of secondary treatment failure in the study cohort. However, it would be useful to explain in further detail how the results of this other study demonstrate evidence of secondary treatment failure rather than simply insufficient dosing or insufficient injection localization. Additionally, this data is missing the reference citation.
· Lines 94-96 seem to state the overall conclusion of the paper. This should be saved for the discussion and conclusion section. Instead, this paragraph should clearly state the aim of the study.
· Aim: Would clarify whether the overarching aim is to determine whether people with cervical and or cranial dystonia who received injections with incobotulinumtoxin A exhibit secondary treatment failure (in which case the drawing of disease severity graphs was just 1 of several methods used to determine whether secondary treatment failure occurred in the cohort) vs specifically whether the drawing of disease severity graphs are useful clinical tools in detecting secondary treatment failure.
Discussion:
· One of the unique aspects of this paper is the use of the CoD graphs. The results section nicely demonstrates that the graphs can be distinguished into 3 distinct curve types (both before and after treatment). However, the significance of these 3 distinct types of graphs is not explored in the discussion section. This is one of the novel aspects of this study and I recommend further discussing the significance of the 3 distinct types of graphs. Does this indicate anything about their disease or predicting their response to treatment? How do these graph types correlate with physician’s scoring of severity? What else can be inferred from these graph subtypes?
· Strengths and limitations: Some additional limitations should be considered/addressed
o Most study participants had cervical dystonia—results may differ for other focal dystonias
o Patient drawn CoD graphs were drawn based on patient’s recall and may introduce recall bias
o Given the lack of primary or secondary failure and lack of comparison to other botulinum toxin formulations, it is difficult to determine whether the study measures were sensitive enough to recognize primary or secondary treatment failure (since there were no positive results for comparison).
Methods:
· Lines 278-279 state that participants with idiopathic cervical dystonia, Meige-syndrome, or oromandibular dystonia were included in the study. However, the results/discussion only seems to mention cervical dystonia. Cervical dystonia does not include or imply cranial or orofacial dystonia. Consistent terms should be used throughout the paper, please clarify.
· It is unclear whether all participants received incobont/A only. Section 1 list this as inclusion criteria, however, lines 298-301 specify criteria for adjustment for comparison of different preparations. Please clarify whether patients may have received other subtypes of botulinum toxin in the past.
· It is unclear whether all evaluations were retrospective or whether further injections were continued and added to graphs after the time of recruitment. Please clarify.
While the overall English language is good, there are numerous spelling, grammatical, and word choice errors which require careful proofreading. Here are some examples from the 1st paragraph:
-Line 51: "di-sulfid" (missing "e" should be disulfide)
-Line 52: "to become biologically activate" should be "biologically active"
-Line 52: this has an end parenthesis but no start parenthesis
-Line 54: "sour milieu" sounds like jargon, consider changing to acidic milieu
Author Response
|
In this manuscript, the authors conducted a study using patient reported trends in disease progression and response to botulinum toxin treatment in addition to testing for neutralizing antibodies and physician impressions of response to determine whether there was evidence of secondary treatment failure in participants with focal dystonia (cervical and/or cranial dystonia). Participants were asked to draw a curve of their disease progression prior to starting treatment with botulinum toxin and a second curve of their progression after initiation of treatment with botulinum toxin. Blood samples taken from each participant at the time of enrollment were tested for the presence of neutralizing antibodies to botulinum toxin. Patient drawn graphs separated into 3 distinct types/categories before botulinum toxin treatment initiation and 3 types/categories after initiation of botulinum toxin. None of the patient samples tested positive for neutralizing antibodies. This was consistent with results on patient drawn graphs of treatment response, which did not show occurrences of either primary or secondary non-response. This paper demonstrates how patient reported measures such as graphing of disease severity can be helpful in demonstrating disease progression and response to treatment. The paper could be strengthened with some clarifications and further revision. Introduction: · Line 65: Sentence “Clinically, the development of NAB-induced may be difficult to detect.” Appears to be missing a word. I believe you meant to say NAB-induced STF · Lines 86-91 the authors present data from another study that demonstrate that more patients reported secondary clinical worsening via CoD graphs than those that had positive MHDA testing. This point is critical, as it serves as the basis for the authors suggestion that their method of drawing the course of disease severity could be more sensitive than MHDA testing in detection of NAB-induced STF and why the CoD graphs were a useful tool to detect evidence of secondary treatment failure in the study cohort. However, it would be useful to explain in further detail how the results of this other study demonstrate evidence of secondary treatment failure rather than simply insufficient dosing or insufficient injection localization. Additionally, this data is missing the reference citation. · Lines 94-96 seem to state the overall conclusion of the paper. This should be saved for the discussion and conclusion section. Instead, this paragraph should clearly state the aim of the study. · Aim: Would clarify whether the overarching aim is to determine whether people with cervical and or cranial dystonia who received injections with incobotulinumtoxin A exhibit secondary treatment failure (in which case the drawing of disease severity graphs was just 1 of several methods used to determine whether secondary treatment failure occurred in the cohort) vs specifically whether the drawing of disease severity graphs are useful clinical tools in detecting secondary treatment failure. Discussion: · One of the unique aspects of this paper is the use of the CoD graphs. The results section nicely demonstrates that the graphs can be distinguished into 3 distinct curve types (both before and after treatment). However, the significance of these 3 distinct types of graphs is not explored in the discussion section. This is one of the novel aspects of this study and I recommend further discussing the significance of the 3 distinct types of graphs. Does this indicate anything about their disease or predicting their response to treatment? How do these graph types correlate with physician’s scoring of severity? What else can be inferred from these graph subtypes? · Strengths and limitations: Some additional limitations should be considered/addressed o Most study participants had cervical dystonia—results may differ for other focal dystonias o Patient drawn CoD graphs were drawn based on patient’s recall and may introduce recall bias o Given the lack of primary or secondary failure and lack of comparison to other botulinum toxin formulations, it is difficult to determine whether the study measures were sensitive enough to recognize primary or secondary treatment failure (since there were no positive results for comparison). Methods: · Lines 278-279 state that participants with idiopathic cervical dystonia, Meige-syndrome, or oromandibular dystonia were included in the study. However, the results/discussion only seems to mention cervical dystonia. Cervical dystonia does not include or imply cranial or orofacial dystonia. Consistent terms should be used throughout the paper, please clarify. · It is unclear whether all participants received incobont/A only. Section 1 list this as inclusion criteria, however, lines 298-301 specify criteria for adjustment for comparison of different preparations. Please clarify whether patients may have received other subtypes of botulinum toxin in the past. · It is unclear whether all evaluations were retrospective or whether further injections were continued and added to graphs after the time of recruitment. Please clarify. Comments on the Quality of English Language While the overall English language is good, there are numerous spelling, grammatical, and word choice errors which require careful proofreading. Here are some examples from the 1st paragraph: -Line 51: "di-sulfid" (missing "e" should be disulfide) -Line 52: "to become biologically activate" should be "biologically active" -Line 52: this has an end parenthesis but no start parenthesis -Line 54: "sour milieu" sounds like jargon, consider changing to acidic milieu |
The authors are thankful to the reviewers for very careful reading and very helpful comments.
This is a nice summary of our manuscript.
Reviewer 1 is right: Correction is performed as suggested.
This point is picked up and is now addressed in detail not only in the introduction but also in the discussion.
The reference is given now.
The aim of the study is now mentioned more clearly.
This is a very helpful comment. The aim of the study is now stated much more clearly.
Reviewer 1 is absolutely right. However, the number of patients is rather small. We therefore have elaborated on this topic in a twice as large study being presented in the same special issue. This study is mentioned as reference [26].
Also in these aspects reviewer 1 is right.
We now address all three aspects explicitly in the strength and limitations section.
We now try to make more clear that patients with CD and patients with other cranial dystonia have been investigated. We now make the distinction between CD and other cranial dystonia.
We are sorry for that confusion. All participants were treated with incoBoNT/A only.
ADOSE, ATSUI, IMPQ, IMPD result from analysis at the day of recruitment. All other parameters were based on retrospective or on a combination of actual and retrospective data.
We checked the manuscript for wrong spelling etc.
This is corrected. This is corrected.
This is corrected.
This is now changed. |

Reviewer 2 Report
Tables 1, 2. “significance-level (p<.05)” And also the p values in the columns. Why not use the full formulation? p<0.05 This also applies to mentions of the parameter p in the text, in particular in line 165.
Line 363. Misprint. “5) . Statistics”
This is a well-planned, lengthy and clearly stated clinical study. I have only one wish, and that is that it seems to me that not enough patients were involved in the study to be absolutely certain of the conclusions. Perhaps the authors can expand their research in the future on their own or with the help of colleagues.
Author Response
|
Tables 1, 2. “significance-level (p<.05)” And also the p values in the columns. Why not use the full formulation? p<0.05 This also applies to mentions of the parameter p in the text, in particular in line 165.
Line 363. Misprint. “5) . Statistics”
This is a well-planned, lengthy and clearly stated clinical study. I have only one wish, and that is that it seems to me that not enough patients were involved in the study to be absolutely certain of the conclusions. Perhaps the authors can expand their research in the future on their own or with the help of colleagues. |
That the p-value is equal to 0.05 is a special situation, which we wanted to address explicitly. In Table 1 and 2 the p-values are now mentioned only when p is smaller than 0.05.
In our version we did not detect a misprint!
Another study on 66 other, consecutively recruited patients with CD is mentioned in reference [25] and [26]. This study will be presented in the same special issue.
|

Round 2
Reviewer 1 Report
The authors have adequately responded to reviewer comments and have appropriately edited the manuscript. The manuscript has been strengthened by the author's revisions.
Author Response
The authors have adequately responded to reviewer comments and have appropriately edited the manuscript. The manuscript has been strengthened by the author's revisions.
Thank you so much for the positive feedback and your valuable comments.